# Effects of Transcranial Electrical Stimulation on Human Auditory Processing and Behavior—A Review

**DOI:** 10.3390/brainsci10080531

**Published:** 2020-08-08

**Authors:** Yao Wang, Limeng Shi, Gaoyuan Dong, Zuoying Zhang, Ruijuan Chen

**Affiliations:** 1School of Life Sciences, Tiangong University, Tianjin 300387, China; wangyao_show@163.com (Y.W.); 17790767651@163.com (L.S.); zore1291428876@163.com (Z.Z.); 2School of Electronics & Information Engineering, Tiangong University, Tianjin 300387, China; 15649871880@163.com; 3School of Precision Instruments and Optoelectronics Engineering Tianjin University, Tianjin University, Tianjin 300072, China

**Keywords:** auditory perception, physiological effects, transcranial alternating current stimulation, transcranial direct current stimulation, transcranial random noise stimulation

## Abstract

Transcranial electrical stimulation (tES) can adjust the membrane potential by applying a weak current on the scalp to change the related nerve activity. In recent years, tES has proven its value in studying the neural processes involved in human behavior. The study of central auditory processes focuses on the analysis of behavioral phenomena, including sound localization, auditory pattern recognition, and auditory discrimination. To our knowledge, studies on the application of tES in the field of hearing and the electrophysiological effects are limited. Therefore, we reviewed the neuromodulatory effect of tES on auditory processing, behavior, and cognitive function and have summarized the physiological effects of tES on the auditory cortex.

## 1. Introduction

The development of neuroimaging technology made it possible to study how brain networks affect behavior and potential cognitive functions [1]. Behavioral experiences can shape the dynamic process of repair and remodeling of remaining neural circuits [2]. Therefore, evaluating how experimentally induced neural changes affect behavior and potential cognitive processes is of utmost importance. Assessment methods in healthy humans mostly involve non-invasive brain stimulation methods.

Transcranial electrical stimulation (tES) is a non-invasive brain stimulation method that uses a low-intensity electrical current to stimulate the target area of the cerebral cortex [3], regulates the excitability of cerebral cortex neurons and the brain wave rhythm [4], and promotes nerve remodeling and repair [5]. TES mainly includes transcranial direct current stimulation (tDCS), transcranial alternating current stimulation (tACS), and transcranial random noise stimulation (tRNS) [6]. In cognitive and clinical treatment, tDCS is probably the most commonly used non-invasive tES method that applies weak currents of different intensities to brain tissue through electrodes (anode, cathode) that are placed on the scalp [7,8]. During tDCS, a constant electric field is created in the brain, which affects the discharge rate of the resting potential of neuronal cells. The effect of tDCS is polarity-related, anode stimulation is related to cell membrane depolarization, and cathode stimulation is related to cell membrane hyperpolarization [9,10,11]. In addition to the constant current, an alternating current can be applied to the scalp through electrodes by applying a specific frequency of the alternating current on the scalp to regulate the oscillation activity of the corresponding frequency band in the brain [12,13,14]. TACS applies a single frequency sinusoidal current [15] to regulate brain oscillations related to physiology at a specific frequency [16]. However, tRNS uses a variety of sinusoidal oscillations of different frequencies, which can generate random amplitudes, and the frequency of the generated noise signal includes all frequencies within 1/2 sampling rate [17]. Stochastic resonance (SR) as a non-linear phenomenon that enhances the detection of weak stimuli or the information content of signals by adding random interference (commonly called noise) [18,19], may be the potential underlying mechanism of tRNS [20].

In recent years, tES has been used to explore the relationship between the activity of specific brain regions and cognition. Moreover, using tES to non-invasively stimulate the brain to regulate neural processes has been extensively studied in the field of brain behavior and has been put into clinical applications [21]. So far, the physiological effects of tES have been the focus of studies in the field of sports [22] and vision [23]; however, little is known about the impact of tES in the auditory domain. Furthermore, there is no accurate conclusion about the electrophysiological mechanism of tES in the field of hearing and its influence on related brain behavior and cognitive function. Therefore, in this study, we review tES-related behavioral and cognitive functions in the auditory field. The PubMed online database was searched using the following keywords in combination to identify articles between 2010 and 2020: transcranial direct current stimulation (tDCS), transcranial alternating current stimulation (tACS), transcranial random noise stimulation (tRNS) combined with auditory electrophysiology, auditory time resolution, auditory attention, and speech. All studies had to be carried out on healthy humans. Figure 1 presents a flowchart with the study inclusion/exclusion criteria.

## 2. Physiological Effects of tES on the Auditory Cortex

Electrophysiological effects of the auditory cortex can be evaluated by analyzing the waveform amplitude of event-related potentials (ERPs) induced by auditory stimuli. ERPs are measured brain responses, which are the direct result of a specific sensory, cognitive, or motor event. In other words, ERPs are stereotypical electrophysiological responses to stimulation, and studying the brain using ERPs provides a non-invasive way to assess brain function [24]. In early studies, Zaehle et al. explored excitability changes in the human auditory cortex after cathodal and anodal tDCS [25]. In their study, an active tDCS electrode was placed in the temporal region (T7) or temporal parietal region (CP5), whereas a reference electrode was placed in the contralateral supraorbital area. Electroencephalography (EEG) electrode placement was performed according to the international 10–20 system [26]. Each subject received four consecutive sessions at one-week intervals, and the experiment lasted one month. In each session, subjects received one sham condition and one active stimulation (anodal or cathodal) condition, in which the sham stimulation condition preceded the active stimulation condition. In the four sessions, two sessions involved stimulation over the left temporal region, including cathodal T7 stimulation and anodal T7 stimulation. The other two stimulations were performed in the left temporal parietal region, and the same cathodal and anodal CP5 stimulation were performed once. This ensured that each participant received each stimulation method, and that the stimulation sequence was balanced. After the stimulation was completed, the auditory evoked potential (AEP) evoked by a 1 kHz sine tone (60-, 70-, 80-, and 90-dBA sound pressure level) with randomly varying intensity was recorded by EEG. AEP is a type of ERP, as a potential response evoked by sound as a stimulus. When a stimulating sound is produced in the outside world, the stimulating sound will induce the auditory system to produce a series of continuous and rapid potential responses [27]. The ability of the brain to regulate the input of sensory stimulation information is one of the normal functions of the brain that is mainly based on inhibition. This function is called sensory gating (SG) [28]. In previous studies, it has been shown that the AEP-P50 component was a new indicator reflecting SG [29]. The AEP-N1 component has multiple sources at the level of the auditory cortex, including the high temporal lobe, and N1 may be related to target recognition in the perception process [30]. Zaehle et al. analyzed the P50 and N1 components related to the early stages of auditory processing [25] to further understand the nature of the neurophysiological mechanism of tDCS in improving auditory clinical related diseases. It was found that the anodal tDCS on the temporal lobe can increase the amplitude of the auditory P50 component, and the cathodal tDCS on the temporo-parietal region can increase the amplitude of the auditory N1 component. Changes in AEP composition indicated the changes in the excitability of the auditory cortex caused by the active tDCS in the temporal lobe and the parietal temporal cortex and may help to understand the underlying mechanism related to the successful treatment of hearing disorders, including tinnitus, via tDCS. According to Zaehle et al., who performed a relatively early tDCS electrophysiological study to change the auditory cortex, Terada et al. continued to study if the application of tDCS in the left dorsolateral prefrontal cortex (DLPFC) can regulate cortical function and evaluated the changes in auditory P50 components as an indicator of SG function [31]. The DLPFC area is considered to be involved in many cognitive processes, such as executive function and SG process [32,33,34]. Subjects in the study performed by Terada et al. underwent stimulation of the left DLPFC area in a random order under stimulation sessions of anodal, cathodal, or sham tDCS [31]. Experimenting on three separate days, the stimulation interval between each single session was one week. The reference electrode was located on the right mastoid. The experiment consisted of four parts: before tDCS, during tDCS, immediately after tDCS, and 30 min after tDCS. In each experimental part, a pair of click auditory stimuli (S1 and S2) was applied for 9 min to induce P50 components. There was a 500 ms interval between the first stimulus S1 and the second stimulus S2, and each experimental part had 100 pairs of click sounds, where tDCS started 6 min before the start of the P50 stimulation period. Throughout the experiment, the EEG of all participants was recorded for approximately 1 h. The results showed that cathodal tDCS significantly changed the average P50 gating index when compared to the other two stimulation conditions. Thus, tDCS may regulate the SG function by reducing the cortical excitability of DLPFC. 

Furthermore, compared to the experimental scheme of Zaehle et al. [25], Kunzelmann et al. investigated the potential effect of tDCS on the excitability of the left posterior temporal cortex [35]. All subjects were randomized to receive anodal stimulation/sham stimulation. Experimenting on two days, the stimulation interval between each single session was one week. The resting state before stimulation, during, and after stimulation was recorded, and the subject was given a passive listening task throughout the recording process. During this passive listening task, the subject listened to a recording containing 400 stimulus tones (loudspeakers presented, 65 dB pure sinusoidal tones). In the experiment, the amplitude and latency of P50, N1, and P200 were analyzed by factors of stimulation conditions (anodal, sham stimulation) and time points (before stimulation, during stimulation, after stimulation). As another component of AEP, P200 appeared about 200 ms after the start of auditory stimulation and may be related to speech processing [36]. When comparing with the results obtained by Zaehle et al., Kunzelmann et al. predicted that the amplitude and latency of the three components would increase after tDCS stimulation. This would further prove the potential modulation effect of tDCS on the excitability of the left posterior temporal cortex and the role of tDCS in auditory processing. Unfortunately, no significant effect of different stimulations on AEP was observed, and there were no significant differences between different time points. Thus, in applications where tDCS regulates auditory processing, it is essential to determine the optimal parameters for the auditory cortex. 

In a recent study, Boroda et al. investigated whether the anodal tDCS targeting the auditory cortex will modulate the plasticity induction of the entire brain area over time [37]. In previous studies, sensory tetanus (ST) paradigms were used as an effective tool to study the mechanism of neural plasticity [38,39]. The characteristics of ST paradigms included a high rate for a brief period that presented one of two sinusoidal tones of different frequencies (1900 Hz and 3000 Hz). During the baseline test, each control tone of sound stimulation lasted 12 min. During auditory ST paradigm stimulation (study pitch for plasticity induction), auditory stimulation lasted for 5 min. TDCS was only conducted during the ST paradigm stimulation. The stimulation site was bilateral primary auditory cortex. The subject received two stimulations (anodal and sham stimulation) with anodes at T7 and T8 and return electrodes at Fp1 and Fp2. With two experimental sessions, the stimulation interval was at least one day. The 45 s after the end of the ST stimulation, the same sound stimulation as the baseline period was required. Then, 30 min after the end of the ST stimulation, the same tone as the baseline tone was started, but only 90 sound stimulations lasting 6 min were presented. The entire experimental process was recorded by EEG for about 60 min. The results showed that tDCS significantly adjusted the plasticity of the study sound compared with the sham stimulation and revealed no effect on the plasticity of the control sound. This effect was time-dependent, and the effect was no longer obvious after 30 min. Taken together, these results provided a strong physiological basis for anodal tDCS to adjust the plasticity of the auditory cortex. Furthermore, the effect of tDCS on the plasticity of the auditory cortex gradually weakened over time. 

In addition to the ERP components mentioned above, the mismatch negativity (MMN) as a negative component of event-related potential (ERP) was also an electrophysiological indicator for automatic processing of auditory information [40]. To explore the effect of tDCS on MMN, in the earliest study by Chen et al., each subject was assessed on four different sessions without tDCS as well as with sham, anodal, and cathodal tDCS [41]. Each experimental session was separated by at least seven days. The total test period was about one month. Depending on the type of stimulation, the anode or cathode electrode was placed on the right frontal cortex (F4), and the reference electrode was placed on the left supraorbital area. After the stimulation, an EEG recording was performed and the MMN auditory stimulation was evaluated. The auditory stimulation consisted of two parts: duration deviation and frequency deviation, and each part was separated by 2 min. The “standard” and “biased” stimuli of MMN in the duration deviation were performed for 50 ms and 100 ms respectively, and the pitch frequency was constant at 333 Hz. In the frequency deviation, the pitch frequencies of the “standard” stimulus and the “deviation” stimulus of MMN were 333 Hz and 353 Hz, respectively, and played at a constant duration of 50 ms. The results showed that the anodal tDCS reduced the frequency amplitude of auditory MMN, however, no changes were observed for the duration deviation. Subsequently, Impey and Knott used small and large pitch deviations under the MMN index to evaluate the effect of tDCS on the subject’s auditory discrimination ability [42]. In brief, subjects underwent two different sessions, including sham and anodal tDCS, with 2–5 days between each session. EEG recording and MMN evaluation were performed before and after stimulation. The auditory stimulation for MMN evaluation included a random mixture of 600 “standard” stimuli and “biased” stimuli. The analysis results of MMN revealed that a relatively lower baseline amplitude and a relatively smaller pitch deviation after anodal tDCS in individuals, and the MMN changes caused by auditory pitch were significantly enhanced [42]. These findings indicated that the more difficult it was to detect abnormal tone changes, the stronger the effect of tDCS on MMN. Furthermore, Impey et al. demonstrated that the anodal tDCS on the auditory cortex can enhance the hearing discrimination ability of healthy subjects [42]. Moreover, anodal stimulation that showed a trend in improving tone discrimination was observed in a study by Impey et al. [43]. 

In the above-mentioned study by Chen et al., it was shown that the anodal tDCS on the right DLPFC can reduce the frequency amplitude of auditory MMN. In addition, Chen et al. attributed this differential effect to the differing networks presumed to be activated by deviations in duration and frequency. However, Weigl et al. thought this might be related to the MMN recording only once after each tDCS stimulation. Therefore, in an additional study, Weigl et al. studied the effect of the left-side DLPFC on the auditory discrimination process [44]. In brief, each subject was assessed during three different sessions, including sham, anodal, and cathodal tDCS on three separate days for a maximum duration of three weeks. Then, 35 min of EEG were recorded before and after stimulation, including 19 min of passive Oddball, 13 min of active Oddball, and 3 min of resting EEG. The Oddball experimental paradigm refers to the continuous alternating presentation of two or more different stimulations, which are divided into standard stimuli and deviation stimuli due to the different probabilities of presentation. Active Oddball used a standard sound and two deviation sounds (see the parameters described by Kipp et al. [45]). Passive Oddball used one standard tone and three deviation tones (frequency deviant, duration deviant, intensity deviant). It was found that the anodal tDCS on the left DLPFC reduced the duration and intensity deviation of auditory MMN but did not affect the frequency deviation. The results revealed that information about the type of deviance was processed in different regions within the prefrontal cortex. Later, in a study by Royal et al., real tDCS or sham stimulation were performed on the right frontal area and right temporal area of subjects in different pitch change detection tasks to study the abnormal pitch detection in these areas [46]. The experiment consisted of three sessions, one for each type of stimulation (frontal, temporal, or sham stimulation). The stimulation interval was one week, and the EEG was recorded 16 min before and 32 min after stimulation. Then, comparative analyses of each electrode, each stimulation type, pitch type (standard, 6.25 cent deviation, 200 cent deviation), and stimulation block (16 min before stimulation, 16 min after stimulation, 16–32 min after stimulation) were conducted. The data showed that cathodal stimulation of the right inferior frontal gyrus (IFG) and auditory cortex (AC) reduced the P3 amplitude induced by small pitch (6.25 cents) deviations compared to sham stimulation. The P3 component was thought to be related with the conscious detection of tone change, to orient attention and working memory processes [47,48]. It was shown that the electrical brain activity related to the detection and consciousness of small pitch deviants was regulated by cathodal tDCS on the IFG and AC, and that using tDCS can change the electrophysiological response of healthy adults to small pitch deviations. The subcomponents of P3, including P3b and P3a, reflect context updating and top-down monitoring of attention. In this study, two different stimulation points affected different subcomponents of P3. Frontal stimulation reduced the P3a component, whereas temporal lobe stimulation reduced the P3b component. These findings supported the view that P3 is related to the ability to detect a deviant tone and provide support for the association between P3 and fronto-temporal brain regions. In addition, these patterns may prompt us to induce the behavioral phenotype of amusia, which requires stimulation at multiple sites, including the frontal and temporal lobes. In summary, combined with the changes in ERP, tES can be used as a potential effective method to identify more auditory-related electrophysiological effects. Thus, to further study the effects of different stimulus positions, it is of utmost importance to identify the correlation of ERP behavior. 

The changes in the auditory cortex induced by tES combined with the auditory steady-state response (ASSR) can be used as another indicator to judge the electrophysiology of hearing. ASSR, as a typical AEP, is usually a follow-up response caused by a continuous amplitude modulation tone or a click trains sequence presented at a given frequency [49]. Synchronous oscillating gamma-band activity (30–100 Hz) has been shown to be a key mechanism for dynamically coordinating across multiple channels [50] and was an important mechanism for integrating sensory information [51]. When sound stimulation was repeated at a rate of about 40 times per second, the phenomenon that brain waves combined to form a single stable waveform was the 40 Hz ASSR [52]. When combined with 20 Hz and 40 Hz-ASSR, the effect of tRNS on auditory cortex-induced activity can be explored. For this purpose, Van Doren et al. used 20 Hz and 40 Hz ASSR to measure the electrophysiological data of the subjects [53]. Each participant was tested in two sessions (real or sham tRNS) in a randomized order with a one-week interval. The first part of the test included 5 min resting EEG and 7 min auditory stimulation EEG. In addition, the auditory stimulus contained 140 repetitions (70 per tone) with a length of 800-ms. At the end of the first part of the measurement, the EEG recording was closed and the real or sham tRNS was started. The second part of the EEG recording was started immediately after the stimulation, including the same auditory stimulation EEG of 7 min and 5 min resting EEG. It has been found that the ASSR caused by FM tones at 40 Hz was increased significantly, thereby indicating that tRNS increased the excitability of the auditory cortex. 

Furthermore, in order to test the effect of tDCS on human gamma band ASSR, Miyagishi et al. tested each participant in two sessions (bilateral real tDCS or sham) in a randomized order, and sessions were separated by one month [54]. The 40 Hz-ASSR was recorded 10 min after each stimulation by magnetoencephalography (MEG). The magnetic field caused by the electrical activity of neurons in the brain was measured using MEG for spatial positioning. During the recording, subjects performed a button task when responding to the 2 kHz click-train stimulus. However, the event-related spectral perturbation (ERSP) and inter-trial phase coherence (ITPC) revealed no significant effect of tDCS on the ASSR of the γ band. These findings were insufficient to prove the effect of tDCS on auditory cortical nerve transmission. Unexpectedly, the data showed enhancement of the ERSP in the β-band of the left motor cortex after tDCS. This finding may be related to the button task, or it may reflect inhibited cortical activity induced by tDCS during the task. Pellegrin et al. found that tDCS inhibited the γ synchronization of the auditory cortex [55]. Because the auditory cortex and hand sensory-motor areas are interconnected [56], Pellegrino et al. studied the effect of tDCS, which targets the primary sensory-motor hand regions on the gamma synchronization of the temporal region away from the stimulation point. Each participant received two sessions (bilateral real tDCS and sham) stimulations (left anodal, right cathodal) at intervals of at least 20 h [55]. Before and after stimulation, MEG was used to measure the γ-synchronization of the cerebral cortex, and auditory stimulation at 40 Hz was performed simultaneously with the MEG measurement. The results revealed that bilateral tDCS over primary sensory-motor hand regions caused a significant reduction in gamma synchronization. Moreover, this effect exhibited significance at the place of most obvious gamma synchronization and covered a large area of the right centro-temporal cortex. The data showed that, to its effect on cortical excitability, tDCS may also be a useful tool for regulating gamma synchronization in addition.

Likewise, Hyvarinen et al. studied the effect of tACS on ASSR changes [57]. To compare the effects of different tACS conditions on auditory-induced steady-state activities. TACS of 12 Hz and 6.5 Hz were performed on the bilateral auditory cortex (BAC) of the subjects. The experimental session included a resting state block (block 1) and five blocks with different combinations of auditory and electrical stimulation. The combined blocks included block 2, which involved 5 min 41 Hz-ASSR; block 3, which involved 5 min 12 Hz-tACS; block 4, which was 5 min 41 Hz-ASSR after 12 Hz-tACS; block 5 which involved 1 min 6.5 Hz-tACS; and block 6, which involved 41 Hz-ASSR after 1 min 6.5 Hz-tACS. Block 1 was started first, the remaining blocks were in random order, and MEG was recorded throughout the entire experiment. The data showed that the power of ASSR was significantly reduced under 12 Hz-tACS conditions. From a physiological point of view, the oscillation activity of the alpha band may inhibit the excitability of the cortex in the corresponding region [58]. Therefore, a possible mechanism of the above-mentioned tACS-induced effect was that 12 Hz-tACS entrained the alpha activity of the auditory cortex itself. Table 1 presents the experimental parameters and results involved in this electrophysiological section.

## 3. The Potential Processing Effect of tES on the Central Auditory Process

### 3.1. The Effect of tES on Auditory Temporal Resolution

The central auditory process involved the auditory system mechanism and process that was responsible for behavioral phenomena, including sound localization and lateralization, the temporal aspect of hearing, and auditory discrimination. The temporal aspects of hearing included temporal resolution, temporal integration, and temporal sequencing [59]. In the process of evaluating time in the auditory field, time resolution played a key role in sound processing in the AC [60]. Initially, Ladeira et al. used the random gap detection test (RGDT) to study the effect of tDCS on auditory processing [61]. The RGDT is a hearing test that recognizes pitch intervals was used to assess the temporal resolution of hearing. Subjects received sham stimulation, cathodal, or anodal tDCS in the BAC, and the reference electrode was placed on the right deltoid muscle. The stimulation interval was 48 h. The 10 min tDCS included 3 min tDCS and 7 min tDCS simultaneous RGDT tasks. In the RGDT test, tones appeared in pairs and the interval between tones increased or decreased from 0 ms to 40 ms (random order). The RGDT task in the experiment included an exercise subtest and four formal subtests with frequencies of 500, 1000, 2000, and 4000 Hz with a length of 7 ms. A randomized test (white noise) of clicks was included in the last randomly appearing subtest, and clicks and tones appeared at intervals of 0, 2, 5, 10, 15, 20, 25, 30, and 40 ms. The subject identified when each pair of tones were separated in time. The detection threshold obtained from the analysis showed that tDCS had a polar dependence on the temporal processing activities of the AC. Compared with the baseline, the time resolution of the anodal area was increased by 22.5%, and the time resolution of the cathodal area was reduced by 54.5%. Furthermore, to test whether the left hemisphere was more advantageous in time processing, Heimrath et al. had subjects undergo sham stimulation, anodal tDCS of the left or right AC within three days [62]. The gap detection task (GDT) was started 10 min after tDCS, and the stimulation continued until participants completed the task. The data showed that the tDCS on the left but not the right AC changed the auditory temporal resolution, reflected left-hemispheric dominance for dealing with rapidly changing acoustic, and that tDCS may become a potential treatment for hearing impairment.

Although the above-mentioned studies showed that anodal tDCS can change the temporal resolution of hearing, the solution to improve the temporal resolution of hearing with tES needs to be further improved. Baltus and Herrmann discovered that a reduction in gamma activity in the AC could reduce the individual’s time resolution [63]. Furthermore, the temporal resolution of the auditory processing mechanism of the cerebral cortex was related to the peak frequency of gamma oscillation (individual gamma frequency, IGF) of neurons in the AC. Individuals with a higher IGF tended to have a better temporal resolution [63]. The GDT task of the human AC was mainly divided into two tasks of the same pure tone (within-channel) and two different pitches (between-channel). In addition, the gap threshold of the between-channel GDT was greater than the gap threshold of within-channel GDT [64]. Therefore, in a study performed by Baltus et al. [65], a finite element model (FEM) modelling was used to identify a tACS electrode device, which was targeted with optimal current orientation for auditory cortical stimulation (one channel per hemisphere: FC5-TP7/P7 and FC6-TP8/P8). In addition, to explore the relationship between the individual frequency of the AC and behavioral performance in the between-channel GDT, Baltus et al. also investigated whether tACS regulation of ongoing gamma activity could improve the performance of the GD task (shorter GD threshold). Therefore, a two-day experiment was initiated, and the EEG analysis of the whole brain electrode was used to identify the electrode with the largest ASSR amplitude (Fz) on the first day. Subsequently, the maximum value of the ASSR was obtained by extracting the fast Fourier’s transform (FFT) value as the estimated IGF value, and the training link of the GD task between channels was performed before and after IGF evaluation. The GD task without tACS stimulation was started the next day. The tACS was performed for 7 min (Group A: tACS frequency above IGF; Group B: tACS frequency below IGF), and GD evaluation was performed again 2 min after the stimulation. The GD task in the experiment was an amplitude modulated tone containing two modulation frequencies (tACS frequency, IGF). Finally, by comparing the GD threshold between groups and the relative changes in ASSR amplitude before and after stimulation, the effect of tACS on oscillatory activity was evaluated. The results showed that the GD performance of Group A with tACS stimulation was significantly improved when compared with that of Group B. The amplitude of the ASSR reached a peak when the first tACS frequency (IGF ± 4 Hz) appeared, however, the natural frequency was restored when the first IGF modulation frequency appeared (IGF value). These findings revealed that the possibility of improving GD by designing tACS’s personalized the stimulation program. In addition, these data also provided further support for the connection between oscillatory activity in the brain and time resolution. 

In a recent study, Baltus et al. combined the above-mentioned experimental ideas, and continued to explore the relationship between IGF and GD performance using tACS in the elderly population [66]. Subjects received individualized tACS at the frequency above IGF 3 Hz (experimental conditions) and below IGF 4 Hz (control conditions), respectively, when performing the task of detecting gaps between channels. It was found that at baseline, IGF was significantly related to the GD performance. In the experimental conditions, tACS modulated GD performance, whereas under control conditions it did not. In addition, among the elderly, the effect of tACS on auditory time resolution seemed to depend on the endogenous frequency of the AC. Specifically, elderly people with lower IGF and auditory time resolution benefited from auditory tACS and showed higher GD performance during stimulation. Therefore, it is necessary to design a personalized tACS scheme for studying the relevant modulation of auditory performance.

Although both tACS and tRNS could regulate auditory gamma activity, tRNS allowed for a frequency unspecific application without prior knowledge on the specific frequency of the target resonator. Rufener et al. assessed the effect on auditory resolution by applying tRNS to the BAC [67]. To evaluate the differences in sound processing between individuals, measurement methods of time resolution and spectral resolution were used, respectively, and involved the inter-channel GDT and the pitch discrimination task (PDT). GDT adjusts the time of the stimulation gap to determine the smallest gap between stimuli that are recognized by the subject. PDT determines the smallest pitch difference that can be detected by the subject by letting the subject identify the higher of the two tones of different frequencies. In brief, subjects received tRNS or sham stimulation in the BAC at an interval of at least six days. Each session performed tRNS or sham for 5 min, then individual GDT and PDT were alternately evaluated for 15 min. After a short break, tRNS was performed again and EEG was recorded after a 5 min break. During the recording, subjects alternately performed two interval tasks and tone tasks. In the interval task, subjects compared a reference sound without intervals and made judgments on three different interval sounds: (I) stimulus significantly higher than the duration of the individual’s sensory threshold interval of 100 ms, representing high signal-to-noise ratio stimulation (SNR_HIGH); (II) a stimulus that was significantly lower than the individual’s sensory threshold interval, and lasted for 20 ms, representing a low signal-to-noise ratio stimulus (SNR_LOW); (III) the interval duration was the stimulation of the assessed individual GDT (critical SNR). Subjects judged whether the sound stimulation included intervals by button. The tone task was divided into 1000 threshold sine tones, individual threshold stimulus (critical SNR) evaluated during PDT, high signal-to-noise ratio stimulus at 1030 Hz (SNR_HIGH), and low signal-to-noise ratio stimulus at 1005 Hz (SNR_LOW). The data showed that tRNS was only useful for detecting gap stimulation at the critical SNR but had no effect on low SNR and high SNR stimulation. Compared with sham stimulation, tRNS only improved the detection rate of near-threshold stimulation in the time domain and had no effect on the discrimination of spectral characteristics. It was shown that the application of tRNS to effectively improve the processing of temporal acoustic features may be achieved by enhancing the endogenous auditory gamma oscillation. Moreover, the critical SNR can be used as an additional indicator of auditory tRNS amplifying the resonance frequency of the auditory system. 

In a previous study, Rufener et al. considered that the left and right hemispheres of the auditory system may have different effects on the processing of auditory features [68]. In their study, subjects were subjected to 1.5 mA tRNS at different positions. While continuously recording the EEG, tRNS stimulation was performed on the left auditory cortex (LAC), right auditory cortex (RAC), or BAC, and sham stimulation was used as a control condition. Subjects had to simultaneously detect an abnormal tone in the three sine tone sequences, while maintaining the stability of all tone durations and standard tones, thereby modulating the interval between tones and the frequency of deviation from the tone. The data showed that the application of tRNS on LAC and BAC was related to the tone detection task, and that LAC had functional relevance when dealing with time features. This effect was modulated by the interval between tones rather than frequency that deviated from the tone. The data showed that tRNS is a technology that can effectively adjust basic auditory perception, and systematically evaluating the most effective tRNS parameters can provide guidance for clinical applications in improving language barriers.

### 3.2. The Effect of tES on Auditory Attention

When individuals were placed in a noisy environment, they could locate the sound source of interest from multiple unrelated sounds and focus their attention on the sound source. The neural basis of a person’s ability to locate sound sources was called the “cocktail-party” effect [69]. To study the potential neural mechanism of a person’s spatial localization of sound sources, Lewald et al. simulated the sound localization in the case of the ‘cocktail-party’ [70], and subjects responded to the source of the target sound by pressing keys. Each stage of the experiment included the following four auditory tasks: (1) immediately before tDCS, (2) during tDCS, (3) immediately after tDCS, and (4) one hour after tDCS. TDCS used two electrode montages (anode left/cathode right; anode right/cathode left) applied to the superior temporal gyrus (STG) or inferior parietal lobule (IPL), respectively. In the control group, somatosensory-motor cortex (SMC) was performed off-target active stimulation. Each stimulation point (STG, IPL, and SMC) was tested by a different group of subjects (both males with normal hearing). The positioning accuracy and error analysis results revealed that the accuracy of target positioning in the left hemisphere can be improved by using the left anode/right cathode bilateral electrodes for superior temporal gyrus, including planum temporale and auditory cortices. In addition, tDCS had no effect on the absolute error of positioning and the percentage of correct responses. It was indicated that the general response accuracy was not regulated by brain polarization, but mainly depended on factors, such as attention or alertness to auditory tasks. These results supported the impact of tDCS on the positioning and spatial separation mechanism of concurrent sound sources. 

To study whether single-dose bihemispheric double-monopolar anodal DC stimulation was more suitable than the study in the previous paragraph to induce improvement of high-order auditory function [71], Lewald et al. simulated the “cocktail-party” situation to locate target words in space. Offline bihemispheric double-monopolar anodal tDCS and sham was performed in the temporal lobe auditory area at intervals of 6–14 days, and the performance of the subjects was evaluated before tDCS, after tDCS, and one hour after tDCS. The results showed that in the simulated “cocktail-party” situation, tDCS applied to superior temporal gyrus, including planum temporale and auditory cortices, improved the accuracy of the positioning of spatial targets. Moreover, compared with sham stimulation, the number of correct positioning after tDCS was increased by an average of 3.7 percentage points, and the accuracy of auditory tasks was only insignificantly reduced after 1 h of stimulation. Therefore, the bilateral tDCS method may become a promising tool for enhancing human auditory attention function. 

Drawing on the experimental ideas of the above-mentioned study, Hanenberg et al. explored the influence of tDCS on the electrophysiological relevant factors of spatial selective attention in hearing [72]. Based on the study by Lewald and Getzmann, who showed that, compared with a single sound source, multiple sound sources resulted in an increase in the amplitude of the P1 component, a decrease in N1 component, and an enhancement in the N2 component. The processes related to selective auditory attention that was most obvious near N2 [73]. Therefore, this study focused on the N2 component in ERP. The experiment was conducted in three weeks, and unipolar sham, anodal, or cathodal tDCS were performed at the right posterior superior temporal cortex. The subjects were divided into two groups, one group included younger subjects (18–30 years old) and the other group included older subjects (66–77 years old). An auditory spatial localization task that simulated the “cocktail-party” situation was completed while recording the ERP. Each stage of the experiment included four blocks—15 min before tDCS, during tDCS, 15 min after tDCS, and one hour after tDCS, and the auditory task was continuously executed for 46 min in the first three blocks. Compared with sham stimulation, ERP data showed that the amplitude of the response of the N2 component on the contralateral (left) rather than the ipsilateral (right) increased significantly by 0.9 μV within 15 min after tDCS. At the same time point, the source location of the cerebral cortex showed a reduction in electrical activity of the posterior parietal cortex (PPC) on the same side (right). In general, behavioral performance was improved in the anodal state, but not in the cathodal or sham stimulation state. These findings were consistent with the above-mentioned related improvement of posterior superior temporal gyrus (pSTG) double-sided unipolar anodal tDCS [71]. Thus, these findings indicate that anodal tDCS can regulate processes related to PPC auditory selective spatial attention function. In addition, although the effect of tDCS on behavioral performance was only demonstrated in younger participants, electrophysiological data showed that DC stimulation had a similar effect in both younger and older subjects. Therefore, the anodal tDCS on the temporal lobe may be effective in improving the auditory function of healthy elderly who decline with age. 

However, when studying the effect of tACS on auditory attention, tACS could correspond to the adjustment of the corresponding frequency oscillation of the cerebral cortex. In previous studies, it was shown that when people focus on a certain position in space, the alpha power difference (8–12 Hz) in the parietal cortex of the cerebral hemisphere would affect the direction of auditory attention to language [74]. Furthermore, the α power, which was produced by neural oscillations in the sensory area of the hemisphere on the same side of the brain was increased, and the contralateral hemisphere was reduced [75,76]. However, γ oscillations were decreased in the same cerebral hemisphere as the attention and were increased in the contralateral cerebral hemisphere. In particular, γ activity was modulated by the phase of the α oscillations [77]. If lateral oscillation was functionally related to spatial attention; then, using tES to stimulate one of the cerebral hemispheres, α and γ oscillations had different regulatory effects on the accuracy of spatial attention. In order to verify these findings, Malte et al. conducted a dichotic listening (DL) task under a continuous tACS (tACS & sham) with a frequency of α (10 Hz) or γ (47 Hz) [78]. The interval between two frequency stimulations was 5–14 days, and in each session, the sham stimulation was performed before tACS stimulation. During the DL task, subjects were asked to only pay attention to the four oral numbers on the left or the right ear (in the range of 21–99 but removed an integer multiple of 10) and ignore the numbers on the other ear. The stimulation position of tACS was left-hemispheric pSTG and the area surrounding the auditory and parietal lobe, and the stimulation electrodes was over FC5 and TP7. It was shown that α-tACS in the cerebral hemisphere suppressed the attention of the target speech located on the opposite side of the stimulus and increased the speech attention on the same side as the stimulus. In addition, it was indicated that transcranially stimulated oscillations can enhance spatial attention and language selection attention. 

Subsequently, Deng et al. studied the control of parietal α-tACS on auditory spatial attention [79]. Two different forms of high definition transcranial alternating current stimulation (HD-tACS) experiments were conducted in the subjects (main experiment: 10-Hz HD-tACS/sham, and the experiment interval was 1–14 days, control experiment: 6-Hz HD-tACS/sham, experiment interval was 1–16 days), and the stimulation position was right intraparietal sulcus (RIPS). In a previous study by Golbarg et al., it was shown that the sudden change of speaker position affected the subject’s top-down spatial attention, thereby disturbing the lateralization of α power [80]. Therefore, in the study by Deng et al., the attention of the participants was guided based on spatial (voice position) or non-spatial tasks (speaker gender). In spatial mission, half of the speakers fixed their speech and position in a continuous test, whereas the other half of the speakers only fixed their position but alternated their speech from one syllable to another in a switching test. In non-spatial tasks, the subjects determined the position of the target sound by looking at the left or right arrow on the screen. By calculating the percentage of correct responses under each attention condition, it was found that the switching experiment performed worse in spatial attention than the continuous experiment in spatial tasks. In the continuous experiment, α-HD-tACS at the RIPS position interfered with the auditory spatial attention of the target on the left, but the θ-HD-tACS had no effect. In addition, parietal lobe stimulation had no effect on auditory attention for non-spatial tasks. 

Furthermore, auditory selective attention could also be evaluated by an oddball paradigm, which would produce P3 components in event-related potentials [47]. The evidence for the P3 component reflecting the neuromodulation locus coeruleus–norepinephrine (LC-NE) system activity was reviewed by Nieuwenhuis et al. [81]. It was also proposed that, when subjects perform auditory tasks, the LC-NE system played a key role in the generation of P3, and that the auditory choice attention may be related to LC-NE and large-scale fronto-parietal cortical network activities. In this regard, Rufener et al. systematically evaluated the role of these subcortical and cortical components in auditory choice attention [82]. Two 30 min tRNS were performed on the left DLPFC position of the subject, with a 10 min break in the middle, then an oddball task was performed during the second stimulation. When the subject detected the target sound, the right index finger was used to perform a key response. EEG recordings were performed during the process, and the same process was performed after three days of sham stimulation. By analyzing the online effect of tRNS under the subjects’ oddball paradigm, it was found that tRNS regulated the excitability of the left DLPFC. In addition, the choice of hearing indicated that there was a type-dependent regulation of stimulation type at the level of behavior and electrophysiology. Compared with the sham stimulation, tRNS reduced the subject’s response time to identify the target sound and fastened up the latency of the P3 component. In summary, tES has a regulatory effect on the spatial attention of hearing, and the stimulation area may connect with the relevant brain area to jointly guide auditory attention. Later, more studies were conducted to pinpoint the brain area involved in auditory attention. In addition, the influence of the relationship between the stimulation effect of tES, the duration of action, and related parameters on the regulation of auditory attention were needed for further studies. Table 2 presents the experimental parameters and results involved in this auditory processing section.

## 4. The Effect of tES on Speech Processing

### 4.1. The Effect of tES on Phoneme Classification

In the process of speech perception, subjects had to recognize and distinguish phonemes. Voice onset time (VOT) describes the time elapsed from the release of closure to the beginning of utterance of a consonant, making a distinction between voiced (/da/) and silent consonants (/ta/) [83]. To study the regulation of speech perception by applying tDCS to the BAC in the speech classification task, Heimrath et al. evaluated the change of VOT of consonant-vowels (CV) syllables (/da/, /ta/) during sham, anodal, and cathodal tDCS conditions [84]. Each stimulation was applied to the BAC at an interval of 48 h. The speech classification task started 10 min after the anodal tDCS or cathodal tDCS while the stimulation was continued. In the speech classification task, subjects completed a synthetic VOT continuum within 1 ms and judged whether the syllable was /da/ or /ta/ to give the key response. The task duration was 12 min. Data analysis showed that AC’s two-sided tDCS had a regulating effect on speech perception. Cathodal tDCS improved the speech classification ability of CV-syllables in a VOT continuum, especially the concurrent cathodal tDCS made the slope curve steeper, thereby indicating that the classification of the two syllables /ta/ and /da/ was more accurate. In the subsequent evaluation of neurophysiological changes after stimulation, subjects performed voice classification tasks while recording EEG to judge whether they heard voiced (/ba/, /da/, /ga/) or unvoiced (/pa/, /ta/, /ka/). The results obtained from these data were consistent with the data presented by Zaehle et al., and showed that after the anodal tDCS, the amplitude of the P50 component of all syllables was increased [25]. The experimental results confirmed the modulation effect of bilateral tDCS on speech perception. 

In addition, studies on neurophysiology showed that there was a relationship between endogenous neural oscillations in the gamma frequency range and phoneme processing [85,86]. As a method that can capture the physiologically related rhythm changes, tACS may be suitable for exploring the functional correlation between gamma oscillation and the acoustic characteristics of speech signals. Therefore, in the study by Rufener et al., 6-Hz and 40-Hz tACS to the BAC of the subjects. Furthermore, the two frequencies of stimulation were separated by 1 week and the control group without tACS [87]. The subjects performed the VOT classification task of the phonemes (/da/ or /ta/) and responded through the left and right mouse buttons when 6 min before the stimulation, 18 min during the stimulation, and 6 min after the stimulation. The experimenter recorded its class marker and response time. By comparing the changes before and after stimulation in the 40 Hz, 6 Hz, and without tACS control groups measured separately, it was revealed that the application of 40 Hz-tACS selectively reduced the improvement of phoneme classification ability compared to the without tACS group and the 6 Hz-tACS group. The results were in line with the data presented in a study by Strueber et al. that showed that 40 Hz-tACS over the occipital-parietal areas resulted in changes in visual motion perception compared to 6 Hz-tACS [88]. In other words, it may be not tACS itself, but the application of task- or feature-related neural oscillations that regulate perception. Therefore, the tACS on the AC can be used to assess characteristic-related oscillation rhythms, and it was believed that this method may have an impact on dyslexics and language processing disorders in the elderly. 

Consequently, Rufener et al. explored the effects of tACS on auditory phoneme processing in the normal elderly population and in young adults [89]. Subjects were randomly received 6 Hz-tACS or 40 Hz-tACS in the BAC, and the two frequencies of stimulation were separated by at least six days. The subjects judged whether /da/ or /ta/ were represented after each stimulation and gave a key response. The experiment started with pre-tACS and the subjects completed the VOT-categorization task first. When the first VOT task ended, the tACS started, and the stimulation continued until the end of the second VOT-categorization run. In each experiment, each VOT-categorization task lasted approximately 8 min. The data indicated that 40 Hz-tACS over the BAC hindered the classification of phonemes in young people and improved the accuracy of phoneme classification tasks in the elderly. Then, it was concluded that the change of γ oscillation was a potential neurophysiological mechanism leading to impaired hearing time resolution in the elderly.

### 4.2. The Effect of tES on the Right Ear Advantage in Speech Processing

Using the DL task, information from the two ears can be integrated into the AC on both sides [90]. In previous studies, it was revealed that the right ear had an advantage during DL in healthy subjects [91], which did not change with tDCS stimulation [92]. Mediated by cortico-cortical callosal fibers [90], the perception of the left ear was related to the increased functional connectivity of the functional gamma-band [93]. Therefore, Jan et al. studied the effect of HD-tACS on selectively regulating the functional connection between cerebral hemispheres and tested the changes in auditory perception of the left and right ears [94]. In brief, subjects received multi-site HD-tACS at 40 Hz bilaterally with a phase lag of 180° and sham stimulation within 1–12 days apart. EEG were recorded to study the DL oscillatory phase dynamics at the source-space level. In the DL task, six paired CV syllables (/ba/, /da/, /ga/; /pa/, /ta/, /ka/) were presented to both ears through closed earphones. The subjects selected the clearest syllables heard through the buttons. The study revealed the characteristics of the oscillating phase at 40 Hz, which reflected the different temporal profiles of the asymmetric phase during the perception process of the left and right ears. In the time window of 36–56 ms, the total average phase asymmetry at 40 Hz of the bilateral BA42 area during left ear perception increased. Moreover, the authors also found that the 180° tACS during DL at group level did not affect the right ear advantage. Follow-up analysis showed that the phase asymmetries inherent in the sham tACS process determined the directionality of behavioral regulation. This indicated that the transfer of speech perception to the left ear was related to the enhancement of the phase asymmetry between the brain hemispheres (close to 180°). In subjects with a low degree of asymmetry (close to 0°), there will be a shift to the perception of the right ear’s speech. The description of the trend of the oscillatory network emphasized the importance of phase-specific gamma-band coupling during the ambiguous auditory perception. In future studies, specific stimulation schemes are needed to solve the differences between individuals due to phase asymmetry. 

Della et al. showed that the basis of the right ear advantage in the DL task was related to the superiority of the left temporal cortex in speech processing [95]. Considering that tRNS had stronger results in some cognitive areas of hearing [96], and the high-frequency transcranial Random Noise Stimulation (HF-tRNS) can enhance the excitability of the cortical area [17], Prete et al. first studied the role of HF-tRNS in speech processing, and discussed the regulation effect of HF-tRNS on right ear advantage [97]. In study, HF-tRNS (100–640 Hz) and sham stimulation were applied to the bilateral (Experiment 1) or unilateral (Experiment 2) AC. The interval between the sham stimulation of experiment 1 and HF-tRNS was at least 2 h. The subjects in experiment 2 received HF-tRNS in the left temporal lobe cortex, HF-tRNS in the right temporal lobe cortex, and sham stimulation, and most of the subjects had three stimulation intervals of one day. The reference electrode was located on the contralateral shoulder, and the tRNS started the DL task after 5 min of stimulation and the subjects selected the clearest syllable by clicking on the screen. In the DL task, dichotic pairs of CV syllables were used as an auditory stimulation and the average duration of CV syllables was about 450 ms. Each syllable consisted of one of 6 stop-consonants /b/, /d/, /g/, /p/, /t/, /k/, and was followed by vowel /a/. These six syllables were paired to produce 30 pairs of time-aligned consonant phonemes, and each pair of consonant phonemes alternated between the left and right ears. The results revealed that the right ear advantage during sham stimulation and HF-tRNS, and the HF-tRNS of the BAC significantly enhanced the right ear advantage effect compared with sham stimulation. The data indicated that bilateral AC HF-tRNS actively adjusted the left hemisphere advantage, while unilateral HF-tRNS did not affect the right ear advantage in the DL mission. Due to the effectiveness of bilateral HF-tRNS to regulate the basic speech processing mechanism, it can be considered for the evaluation and treatment of language disorders. Furthermore, it was demonstrated that it is necessary to compare different technologies, montage size and position, and stimulation parameters to clarify their advantages and limitations. Table 3 presents the experimental parameters, results, and auditory paradigm involved in tES effects in syllables classification and right ear advantage section.

### 4.3. Regulation of tES on Speech Understanding

A person’s understanding of speech depends on the spectral content of the speech signal and the integrity of the time envelope. Speech has inherent time characteristics, and the low-frequency information in the 4–8 Hz range transmitted by its amplitude envelope provides the basis for the prosodic hierarchy of the spoken structure [98].

The level of speech understanding is related to the similarity between the time envelope frequency and the cortical activity of the subjects (stimulus-cortex frequency matching) and the phase lock between the two time envelopes (stimulus-cortex phase locking) [99]. This entrained cortical activity between the inherent oscillation envelope of the cerebral cortex and the time envelope of the ongoing speech task is crucial for speech understanding. One study showed that 10 Hz-tDCS has a regulatory effect on auditory sensitivity [100], likewise, Lars et al. found that under the condition of 4 Hz-tACS, changes in the relative time of sound stimulation and electrical stimulation affected the corresponding auditory perception [101]. Therefore, Wilsch et al. considered that the degree of understanding of speech can be adjusted by tACS [102]. For this purpose, each subject completed two phases of the experiment on two different days in their study. The first phase was to confirm the subject’s tACS threshold and the hearing threshold of each ear, and in the second phase, individuals began to familiarize with Oldenburger sentence test (OLSa) process. After being familiar with the process, participants completed the 9 OLSa under 6 tACS time lags, which were randomly assigned, and three control conditions. The tACS time lags (0–250 ms, in 50-ms step) were defined as the time point of tACS stimulation after the onset of the speech signal. OLSa was used to obtain the speech comprehension threshold (SCT) in noise. In the study, nine test lists were selected for OLSa, and each list had 30 sentences composed of five words that were separated by self-paced breaks between tests. The stimulation electrodes of tACS were placed on Cz and bilateral primary auditory cortex T3 and T4. As three control conditions of tACS, anodal, and cathodal direct current (DC+, DC−) matched to the duration of the speech signal, or sham stimulation matched to the duration of the speech signal. The stimulation signal corresponded to the envelope of the concurrent speech signal was called envelope tACS. The envelope of each OLSa sentence was extracted and the envelope-tACS was generated. In the end, each sentence was presented in a combination of noise and envelope-tACS. To analyze the modulation of the speech understanding degree under different tACS lag time, a single subject sine, linear, and quadratic fitting analysis were performed. Subsequently, it was shown that the sinusoidal fit with an average frequency of 5.12 Hz has a good effect on the envelope-tACS modulation of speech intelligibility, and the average frequency of the sinusoidal fit (5.12 Hz) corresponded to the peak of the power spectrum of the speech envelope. Combined, the data showed that sinusoid envelope-tACS adjusted the speech intelligibility in noise. In addition, tACS with a 100-ms lag time may be more effective than other lag times, however, this parameter showed a big difference between different participants.

Since the target speech in this experiment was disturbed by background noise, Zoefel et al. hypothesized that the results were likely due to tACS improved audition scene analysis [103]. Likewise, in a previous study by Riecke et al., it was shown that when the oscillating rhythm of the target speech was synchronized with the endogenous δ/θ neuron oscillation induced by tACS, the target speech was recognized faster in the noisy background, rather than tACS adjusted the understanding of the speech directly [104]. Therefore, Zoefel et al. improved the experiment to ensure that the effect of tACS was to adjust the speech understanding itself [103]. In the study, subjects received different electrical stimulation parameters in two experiments, whereas all other parameters were the same. In experiment 1, unilateral (left hemisphere) stimulation was used, the electrode position was T7 and C3. Experiment 2 used bilateral stimulation over T7 and T8, and the traditional rectangular electrode was changed for a ring electrode. Two experiments were performed twice in the order of ‘no stimulation (A)-tACS-sham-sham-tACS-no stimulation (A)-sham-tACS-tACS-sham’ cycle. During this process, the voice task was performed simultaneously, and consisted of five-words with noise-vocoded, which spoke words according to the rhythm of the metronome, so that the “perceptual centers” or “p-centers” of the voice were consistent with the rhythm of the metronome to obtain a regular voice sequence. During the silent period, participants selected pictures matching to the second, third, and fourth word on the screen they heard. If the participants did not make a choice within the specified time interval, the task automatically continued. Then, the effect of eight phase relationships (between 0 and 7π/4, in steps of π/4; corresponding to delays between 0 ms and 280 ms, in steps of 40 ms for speech at 3.125 Hz) between tACS and speech were assessed by continuously applying tACS and changing the speech presentation’s time, so that the “p-centers” of the individual word would be consistent with a certain tACS phase. The results revealed that tACS induced speech perception adjustment and increased the accuracy of syllable judgment. However, this adjustment can only be confirmed in the case of bipolar stimulation using ring electrodes (instead of unilateral left hemispheric stimulation using square electrodes). Moreover, the phase change between the bilateral tACS and speech from optimal to non-optimal caused the accuracy of the speech report to change by about 8%. 

The impact of electrical stimulation on the degree of speech understanding may also be related to the phase relationship between the envelopes of the corresponding speech, and the interaction within the lag time. Kadir et al. studied the effect of the interaction of the phase and lag time on speech understanding [105]. In brief, subjects were tested for speech interference with synchronous tACS. The stimulation electrodes of tACS were located on T7 and T8 of the AC, and two reference electrodes were placed on both sides of Cz. In the simultaneous language interference experiment, subjects repeated a target sentence (female voice) they heard in the background noise (composed of four male voices), and their answers and score were recorded. The SNR of the sentence was changed through an adaptive process to estimate the SNR corresponding to the sentence reception threshold (SRT) at which the subject could understand 50% of the target voice. The sham stimulation was used as a control condition, and the direct current of the cathodal and anodal was used as an additional control condition and stimulated with the envelope of an unrelated sentence. The stimulation current of tACS corresponded to the speech envelope, but six different phase changes were made at two fixed lag times of 100 ms and 250 ms. It was found that only the sinusoidally changing signal had a phase change equal to the time change by analyzing the relationship between the phase of the voice envelope and the time change. Due to the non-sinusoidal speech envelope, it was ensured that the phase-shifted speech envelope was independent of the time offset of 100 ms and 250 ms. There was a significant phase-dependent difference (120° and 360°) in the SRT at 100 ms and 250 ms lag time points. For a lag time of 100 ms, a phase shift of about 240° produced the best SRT, and a phase shift at about 0° produced the worst SRT. Moreover, for a 250 ms lag time, the best SRT occurred at 60°, and the worst SRT occurred at 0°. In addition, only in the optimal combination of phase and lag time, envelope-tACS with speech envelope can improve speech understanding. However, the combination of current stimulation signal envelope and unrelated speech signal envelope will result in a decline in speech understanding. In summary, for the envelope of non-sinusoidal speech, the phase change and time change of the current stimulation could adjust the speech understanding in different ways. The different adjustments of speech understanding by two different lag times may reflect its different roles in speech processing.

In a recent study, Keshavarzi et al. combined tACS and speech understanding tasks, and studied the effect of delta (δ) and theta (θ) frequency band tACS on speech understanding [106]. The anode electrode of tACS was placed on T7 and T8 of the AC, and the cathode electrode was placed on both sides of Cz. Participants received sham stimulation at the beginning of each sentence, and the SNR was measured through the subject’s adaptive procedure changes when the speech understanding was 50%, which was the SRT value. Then, the subject’s language comprehension ability was measured under 15 different tACS waveforms. These 15 waveforms included one waveform used to provide sham stimulation, the envelopes of the δ and θ bands were shifted by six different phases respectively to produce 12 types of waveforms, and the δ and θ waveforms were obtained from an unrelated sentence. For each waveform, each subject was provided with 25 sentences of speech-shape noise at the SNR corresponding to the previously measured individual SRT, and current stimulation was applied. Participants repeated the content of the sentences they heard, and at the same time recorded and scored the repeated content. During the test period, a total of 375 sentences were performed on two different days. The envelopes of the two bands of δ and θ shifted six different phases of 0°, 60°, 120°, 180°, 240°, and 300° respectively, to study the modulation effect of neurostimulation of these two bands on language understanding. It was found that tACS affected speech understanding in the θ band but had no effect on speech understanding in the δ band. In addition, compared with the sham stimulation, the current stimulation in the θ band improved the understanding of speech in noise, thereby indicating that the θ wave played a significant role in the modulation of speech understanding. Although this method did not have a lag time between speech and envelope-based current, the data showed that to some extent, the phase shift used in the experiment was related to the lag time of approximately 200 ms. Unlike previous studies [102], the phase change of different subjects in this experiment had the same effect on language understanding. The variability in the previous results may be related to the voice used in the study being changed to a single rhythm. Table 4 presents the experimental parameters, results and auditory paradigm involved in tES effects in the speech understanding section.

In summary, tACS can adjust the understanding of speech. The setting parameters of different tACS and the type and position of electrodes will influence the experimental results. Therefore, the design of tACS needs to be further optimized to achieve the best voice modulation effect. In addition, the phase relationship of the best or worst speech accuracy is different within different subjects, and currently, it is not possible to explain these differences. In the future, conducting further research in combination with the EEG/EMG method may be possible. 

## 5. Outline and Future Perspectives

In this study, we outline the research of tES in the field of behavior and cognition around the three aspects of tES auditory electrophysiological effects, auditory processing, and speech processing. In the electrophysiology study, tES was combined with EEG or MEG to better understand the nature of neurophysiological mechanisms through analyzing ERP components. Importantly, tES data collection will produce significant artifacts; therefore, there are many experiments to study the effect of offline tES on auditory perception. Electrophysiological data needs to be measured offline after the stimulation is terminated [25,41]. Therefore, additional experiments were needed to confirm the impact of tES on potential neural mechanisms. In addition, the measurement of tES electrophysiology helps to understand the clinical research related to auditory processing in different diseases, including neurological diseases [107,108], bipolar disorder [109], and aphasia [110], among others.

Subsequently, we described the impact of tES on auditory processing in terms of auditory temporal resolution and selective spatial attention. In previous studies, a connection was shown between the frequency of auditory endogenous neuron oscillation activity and the temporal resolution of perception [63]. As a method of adjusting the frequency of neural oscillations, tACS affects endogenous brain oscillations by changing the frequency of brain oscillations, which may be effective in improving the auditory function in healthy elderly people who have a lower frequency of endogenous oscillations due to age [111,112]. In addition, auditory spatial attention studies aim to improve the auditory cognitive function of healthy people with a lower frequency of endogenous oscillations due to age. In general, the complex cognitive functions are mediated by multiple functionally connected brain regions, and the impact of tES on the corresponding brain activity is a goal of the study [113]. Therefore, additional experiments were needed to combine tES and neuroimaging technology to explore the connection between different areas of the auditory network [114].

Speech processing deficit was one of the most dominant cognitive symptoms of dyslexia [115,116]. Impaired auditory processing reduces the ability to accurately segment the speech stream into its important speech components (such as rhymes, syllables, and phonemes). Therefore, regarding studies on the influence of tES on speech processing, we summarized the influence of tES on cv-syllable classification and speech understanding. So far, tDCS has achieved positive results in some cognitive areas related to speech tasks [117,118]. The potential of tACS to induce neural oscillations at different frequencies in AC seems to be effective in restoring the altered oscillation patterns of patients with dyslexia [87,89,119]. However, few studies have been published on the improvement of auditory and language perception by tRNS [97]. Considering that, in some cognitive fields, tRNS may be more effective than tDCS [96], it is necessary to explore the role of tRNS in speech processing.

Regarding the effects of three different types of tES on auditory behavior, and considering the physiological and neuromodulation characteristics of tDCS, the behavioral effects produced by tDCS may be related to the adjustment ability of task-related neural processing rather than the temporal and spatial specificity of the electric fields generated by the stimulation itself [120]. Therefore, the connection between tDCS and auditory behavior will be manifested by the changes in ERP amplitude induced by different auditory tasks [31,46]. However, due to the lower temporal resolution of tDCS, tACS was used in more cases to explore changes in behavior within a higher temporal resolution. TACS promoted neuronal activity in specific frequency bands [121,122] to study the causal connection between brain rhythm and specific aspects of auditory behavior [57,100,101]. However, there is still some controversy about the related oscillation mechanism of tACS affecting behavior. To study how tACS carries or regulates oscillatory activity in the auditory cortex, it is also necessary to use a multimodal non-invasive brain stimulation recording technology and development reliable artifact rejection methods to identify neural oscillations during stimulation [123,124]. In addition, the tRNS method focuses on studying the relationship between behavior and specific frequency noise inherent in neural processing [17]. The combined application of tRNS and cognitive tasks can further elucidate the relationship between brain oscillations and auditory behavior [67,68]. In addition, tRNS may be more effective than tACS and tDCS in improving human behavior [96,125,126]; therefore, it is necessary to further explore the role of tRNS in speech processing. 

In summary, the different ways of applying the tES method are suitable for studying the interaction between different types of physiology and behavior or cognition. Although some conclusions about the mechanism of tES on the auditory cortex and its impact on auditory cognition have direct electrophysiological evidence, clear conclusions are still needed to draw. Therefore, it is crucial to further study the potential role and effect of tES in the auditory field. Electrophysiological recording can further be combined with tES to clarify the relevant neurophysiological mechanisms involved. In addition, because the tES method is relatively new, the experimental scheme has not yet reached optimal standardization, therefore, it is necessary to consider different mechanisms of action to select the optimal design for the stimulation experiment. Furthermore, the effects of stimulation parameters, electrode position and shape also need to be further discussed. It has been proven that tES is clinically effective in treating hearing-related diseases, therefore, to further explore the long-term effects of tES is clearly warranted. 

## Figures and Tables

**Figure 1 brainsci-10-00531-f001:**
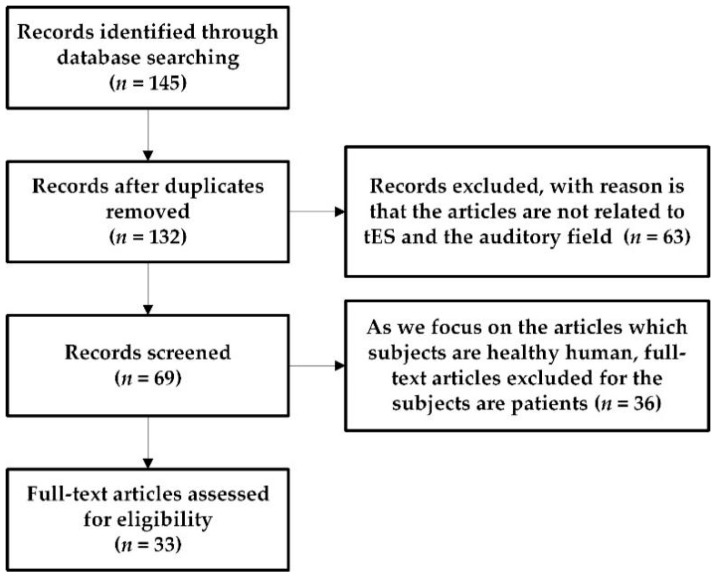
A flowchart presenting the study inclusion/exclusion criteria. We investigated a total of 145 articles in the database, excluding 13 duplicate articles. Further rough screening excluded 63 articles that the articles are not related to transcranial electrical stimulation (tES) and the auditory field. Finally, 33 articles were selected for review.

**Table 1 brainsci-10-00531-t001:** Partial parameters and results of electrophysiological studies.

Reference	Sample Size	Stimulation Types	Stimulation Electrode Position and Size	Reference Electrode Position and Size	Stimulation Parameters	The Number of Stimulation Sessions	EEG Recording Time	Results
Zaehle, et al., 2011 [25]	14	tDCS: anodal/cathodal/sham	T7 or CP5, 35 cm^2^	Contralateral supraorbital, 35 cm^2^	1.25 mA for 11 min.	Each participant received four sessions, each session included one sham and one active tDCS	Nine minutes after real and sham stimulation;	Anodal stimulation of the temporal lobe increases the amplitude of auditory P50, cathodal stimulation of the temporo-parietal region increases the amplitude of N1
Terada, et al., 2015 [31]	10	tDCS: anodal/cathodal/sham	F3, 35 cm^2^	right mastoid, 35 cm^2^	1.0 mA for 15 min	Three stimulation sessions	The entire experiment process is about 60 min	Cathodal tDCS significantly changed auditory P50 sensory gating index Feeling gating may be regulated by the cathodal tDCS of the left DLPFC, not by the anodal/sham tDCS
Kunzelmann, et al., 2018 [35]	24	tDCS: anodal/sham	over TP7 and P7, 35 cm^2^	over Fp2, AF4, and AF8 25 cm^2^	1.0 mA for 20 min	Two stimulation sessions	The entire experiment process is about 60 min	Different stimulations have no significant effect on AEP, and there is no significant difference between different time points Electrode montage location, stimulation intensity, duration, and different stimulation conditions may have a significant influence on the experimental results
Boroda, et al., 2020 [37]	22	tDCS: anodal/sham	T7 and T8, 3.14 cm^2^ PiStim electrode	Fp1 and Fp2 3.14 cm^2^ PiStim electrode	1.0 mA for 5 min	Two stimulation sessions	The entire experiment process is about 60 min	Regardless of tDCS, during the target pitch, the N100 component is enhanced compared to the baseline, and there is no difference during the control pitch Compared with sham stimulation, tDCS regulates the plasticity of the target sound
Chen, et al., 2014 [41]	10	tDCS: anodal/cathodal/sham/no stimulation	F4, 35 cm^2^	Left supraorbital unspecified size	2 mA for 25 min.	Four stimulation sessions	19 min after stimulation.	The anodal tDCS on the right DLPFC reduced the frequency deviation of auditory MMN, but there was no change for the duration deviation
Weigl, et al., 2016 [44]	18	tDCS: anodal/cathodal/sham	F3, 35 cm^2^	right supraorbital 35 cm^2^	1 mA for 15 min	Three stimulation sessions	Thirty-five minutes before and after stimulation	The anodal tDCS on the left DLPFC can reduce the time course and intensity deviation of auditory MMN, but does not affect the frequency deviation
Impey and Knott 2015 [42]	12	tDCS: anodal/sham	C5, T7, unspecified size	Contralateral forehead, unspecified size	2 mA for 20 min	Two stimulation sessions	Unspecified	After the anodal tDCS, the MMN amplitude caused by the small tone is increased The anodal tDCS on the auditory cortex can increase the hearing discrimination ability of healthy subjects
Royal, et al., 2018 [46]	14	tDCS: frontal, temporal, sham	between electrodes Fp1, AF3 and AF7, 35 cm^2^	(1) frontal: between the AF8 and F8, (2) temporal: between the T8 and TP8, (3) Sham: random, 35 cm^2^	2 mA for 20 min	Three stimulation sessions	16 min before stimulation and 32 min after stimulation	The amplitude of P3 in the frontal and temporal regions after cathodal stimulation was lower than the baseline before stimulation The areas around tDCS inferior frontal gyrus and auditory cortex can induce temporary changes in brain activity induced when dealing with pitch deviations
Van Doren, et al., 2014 [53]	14	tRNS/sham	Simultaneous T7 and T8; 35 cm^2^	unspecified	2 mA for 20 min	Two stimulation sessions	Twelve minutes before and after stimulation.	The ASSR caused by the frequency modulation tone of 40 Hz has increased significantly TRNS can induce increased excitability in the auditory cortex
Miyagishi, et al., 2018 [54]	24	tDCS: left anodal, right cathodal/sham	Simultaneous F3 and F4; 35 cm^2^	unspecified	2 mA for 13 min	Two stimulation sessions	MEG test method	tDCS has no significant effect on the ASSR in the γ band Enhanced β-band ERSP of the left motor cortex after tDCS
Pellegrino, et al., 2019 [55]	15	tDCS: left anodal, right cathodal/sham	Simultaneous C3 and C4, 35 cm^2^	unspecified	2 mA for 20 min	Two stimulation sessions	MEG test method	Since the most affected area is far away from the stimulation point, tDCS has a strong remote effect tDCS will significantly reduce the γ synchronization of the auditory cortex
Hyvarinen, et al., 2018 [57]	18	Sine 12 Hz-tACS, 6.5 Hz-tACS no stimulation	Simultaneous T3 and T4, 35 cm^2^	unspecified	(1) no stimulation:10 min (2)12 Hz-tACS: 1.5 mA for 10 min; (3) 6.5 Hz-tACS: 1.5 mA 2 min	6.5 Hz-tACS and 12 Hz-tACS twice each.	MEG test method	Compared with no stimulation, the power of ASSR is significantly reduced under the tACS condition of 12 Hz, while the ASSR of 6.5 Hz-tACS is not reduced

**Table 2 brainsci-10-00531-t002:** Partial parameters and results of auditory process studies.

Reference	Sample Size	Stimulation Types	Stimulation Position and Electrode Size	Reference Electrode Position and Size	Stimulation Parameters	The Number of Stimulation Sessions	EEG Recording Time	Auditory Paradigm	Results
Ladeira, et al., 2011 [61]	11	tDCS: anodal/cathodal/sham	Simultaneous T3 and T4, 35 cm^2^	Contralateral deltoid muscle 35 cm^2^	2.0 mA for 10 min	Three stimulation sessions	unspecified	Random gap detection test	Anodal tDCS enhanced temporal resolution and cathodal diminished temporal resolution
Heimrath, et al., 2014 [62]	15	tDCS: anodal/sham	T7, T8, 25 cm^2^	Contralateral C4, C3, 50 cm^2^	1.5 mA for 13 min	Two stimulation sessions	unspecified	Between channel gap detection task	Anodal tDCS over the left auditory cortex diminished temporal resolution
Baltus, et al., 2018 [65]	26	IGF ± 4 Hz tACS	Left hemispheric: FC5 and TP7/P7 Right hemispheric: FC6 and TP8/P8 Each hemispheric two round electrodes (2.5 cm diameter)	unspecified	1.0 mA for 7 min	Each participant had one stimulation session within two days	Day 1: 40 min in the IGF estimation. Day 2: 1.5 min in the aftereffect estimation.	Between channel gap detection task	Participants in Group A (tACS frequency above IGF) performed significantly better during tACS compared to participants in Group B (tACS frequency below IGF) Both groups had significant relative changes in the amplitude of ASSR
Baltus, et al., 2020 [66]	16	IGF + 3 Hz tACS or IGF − 4 Hz tACS	Left hemispheric: FC5 and TP7/P7 Right hemispheric: FC4 and TP8/P8 Each hemispheric two round electrodes (2.5 cm diameter)	unspecified	1.0 mA for 8.5 ± 1.7 min	Each participant had two stimulation sessions within two days	Day 1: 20 min in the IGF estimation.	Between channel gap detection task	The IGF was significantly related to the gap detection performance under the condition that the baseline and tACS frequency were higher than IGF 3 Hz Under the condition that the tACS frequency was higher than IGF 3 Hz, tACS could modulated gap detection performance
Rufener, et al., 2017a [67]	20	HF-tRNS (100–640 Hz)/sham	Over T7 and T8 35 cm^2^	unspecified	1.5 mA for 40 min	Two stimulation sessions. Stimulation twice in each session.	Record EEG during gap task and pitch task. Unspecified time.	Between channel gap detection task and pitch discrimination task.	Auditory tRNS only increased the detection rate in the temporal domain, no such effect on the discrimination of spectral features The facilitation effect of tRNS was limited to the processing of stimuli near the threshold
Rufener, et al., 2017b [68]	18	tRNS/sham unspecified frequency	LAC, RAC, BAC unspecified size	unspecified	1.5 mA for 30 min	Four stimulation sessions.	tRNS simultaneous EEG About 30 min	Between channel gap detection task and pitch discrimination task.	The application of tRNS on LAC and BAC was related to task performance LAC had functional relevance when dealing with temporal features, and this effect was modulated by the the inter-tone interval between the tone triplets
Lewald 2016 [70]	74	tDCS: anodal and cathodal/sham	(1) superior temporal gyrus (STG, *n* = 24) (2) inferior parietal lobule (IPL, *n* = 28) (3) somatosensory-motor cortex (SMC, *n* = 22). Two round electrodes (diameter 21 mm, 3.5 cm^2^)	unspecified	0.4 mA for 12 min	Each participant had one stimulation session at a different stimulation site	unspecified	Simulated “cocktail-party” situation	The tDCS over superior temporal gyrus could enhance the accuracy of target localization in left hemispace TDCS over IPL and off-target active stimulation over SMC found no significant effects
Lewald 2019 [71]	24	tDCS: anodal/sham	Midpoint between C5 and T7 locations and C6 and T8 locations 35 cm^2^	over the contralateral shoulders 7 × 14 cm	1.0 mA for 32 min	Two stimulation sessions	unspecified	Simulated “cocktail-party” situation	TDCS over the region of posterior STG, significantly improved the performance in localizing a target speaker Compared with sham tDCS, the average increase in correct responses by 3.7% after active tDCS
Hanenberg, et al., 2019 [72]	39	tDCS: anodal/cathodal/sham	Midpoint between C6 and T8 35 cm^2^	over the contralateral shoulder(left) 7 × 14 cm	1.0 mA for 16 min	Three stimulation sessions.	Whole session about 91 min;	Simulated “cocktail-party” situation	The amplitude of the response of the N2 component on the contralateral (left) increased significantly within 15 min after tDCS At the same point in time, the posterior parietal cortex (PPC) electrical activity reduced on the same side (right)
Malte, et al., 2018 [78]	20	10 Hz tACS and sham or 47 Hz tACS and sham	FC5 and TP7 Round electrodes of 3 cm diameter	unspecified	1.0 mA for 25 min	Each participant received two sessions, each session included one sham and one real tACS	unspecified	Dichotic listening task	Unihemispheric α-tACS relatively reduced the recall of targets contralateral to stimulation, increased recall of ipsilateral targets Contrary to α-tACS, γ-tACS relatively increased the recall of targets contralateral to stimulation, decreased recall of ipsilateral targets
Deng, et al., 2019 [79]	38	10 Hz HD-tACS/6 Hz HD-tACS/sham	P2 ring electrode unspecified size	CP2, P4, Pz, PO4 ring electrode unspecified size	1.5 mA for 20 min	Each participant received one sham and one HD-tACS	unspecified	Selective auditory attention task	RIPS’ α-HD-tACS disrupts auditory spatial attention for leftward No effect on performance in the myriad control conditions (i.e., gender, continuous/switching experiment, theta stimulation)
Rufener, et al., 2018 [82]	20	HF-tRNS (100–640 Hz)/sham	F3 25 cm^2^	right shoulder 35 cm^2^	1.5 mA for 30 min executed twice per session	Two stimulation sessions.	tRNS simultaneous EEG About 60 min	Oddball paradigm (target tone recognition)	TRNS regulates the excitability of the left DLPFC TRNS reduced the subject’s response time to identify the target sound, and fastened up the latency of the P3 component

**Table 3 brainsci-10-00531-t003:** Partial parameters, paradigm and results of tES effects in syllables classification and right ear advantage.

Reference	Sample Size	Stimulation Types	Stimulation Position and Electrode Size	Reference Electrode Position and Size	Stimulation Parameters	The Number of Stimulation Sessions	EEG Recording Time	Auditory Paradigm	CV Syllables	Results
Heimrath et al., 2016 [84]	13	tDCS: anodal/cathodal/sham	Simultaneous T7 and T8, 25 cm^2^	Cz 50 cm^2^	1.5 mA for 22 min.	Three stimulation sessions	Record EEG during CV-task II. Unspecified time.	Phoneme categorization task (CV-task I, II)	I: /da/ or /ta/ II: /ba/, /da/, /ga/ or /pa/, /ta/, /ka/	Cathodal tDCS improved the speech classification ability of CV-syllables in a VOT continuum After the anodal tDCS, the amplitude of the P50 component of all syllables increased
Rufener et al., 2016a [87]	Group 1: *n* = 21 Group 2: *n* = 17	Group 1: 6 Hz-tACS/40 Hz-tACS Group 2: no stimulation	Simultaneous T7 and T8, 35 cm^2^	Unspecified	1.0 mA for 18 min	Two stimulation sessions	Unspecified	Phoneme categorization task (CV-task).	/da/ or /ta/	Application of 40 Hz-tACS selectively reduced the repetition-induced improvement of phoneme categorization abilities
Rufener et al., 2016b [89]	Young subjects: *n* = 25 Older subjects: *n* = 20	6 Hz-tACS/40 Hz-tACS	Simultaneous T7 and T8, 35 cm^2^	Unspecified	1.0 mA for 8 min	Two stimulation sessions	Unspecified	Phoneme categorization task (CV-task).	/da/ or /ta/	40-Hz tACS on the bilateral auditory cortex, it was diminished task accuracy in young adults, older adults improved the accuracy of phonetic categorization
Jan et al., 2019 [94]	26	40 Hz-HD-tACS/sham	Bilateral AC eight Ag/AgCl electrodes 12 mm diameter	Unspecified	1.0 mA for 20 min	Two stimulation sessions	5 min before and after stimulation;	Phoneme categorization task (CV-task)	Distinguish between /ba/, /da/, /ga/, /pa/, /ta/, /ka/	The 180° tACS during DL at group level did not affect the right ear advantage The phase asymme-tries inherent in the sham-tACS process determined the directionality of behavioral modulations
Prete et al., 2018 [97]	Group 1: *n* = 50 Group 2: *n* = 24	Group 1: bilateral HF-tRNS and sham Group 2: unilateral HF-tRNS and sham	Group 1: Simultaneous T3 and T4. 5 × 9.5 cm^2^ and 5 × 5 cm^2^ Group 2: T3, T4. 5 × 5 cm^2^	Group 1: unspecified Group 2: contralateral shoulder 5 × 9.5 cm^2^	1.5 mA for 20 min	Two stimulation sessions	Unspecified	Dichotic pairs of CV syllables	Pairs /ba/, /da/, /ga/, /pa/, /ta/, /ka/	The right ear advantage during sham-stimulation and HF-tRNS The HF-tRNS of the bilateral AC was significantly enhanced the right ear advantage effect compared with the sham stimulation

**Table 4 brainsci-10-00531-t004:** Partial parameters, paradigm and results of tES effect speech understanding studies.

Reference	Sample Size	Stimulation Types	Stimulation Electrode Position and Size	Reference Electrode Position and Size	Stimulation Parameters	The Number of Stimulation Sessions	Auditory Paradigm	TACS Time Lags	TACS Phase Relationships	Results
Wilsch, et al., 2018 [102]	19	Envelope—tACS/sham	Simultaneous Cz (35 cm^2^) T3 and T4 (4.18 × 4.18 cm)	Unspecified	between 0.4 mA–1.5 mA unspecified time	Two stimulation sessions	Oldenburg sentence test	0–250 ms, in 50-ms steps	Unspecified	The sinusoidal fit with an average frequency of 5.12 Hz has a good effect on envelope-tACS modulation of speech intelligibility
Zoefel, et al., 2020 [103]	Group 1: *n* = 27 Group 2: *n* = 19	Group 1: unilateral tACS and sham Group 2: bilateral tACS and sham	Group 1: T7 (3 × 3 cm) C3 (5 × 7cm). Group 2: T7 and T8 doughnut-shaped electrodes inner ring (d ^1^ = 2 cm, t ^2^ = 0.1 cm) external ring (outer d ^1^ = 10 cm, inner d ^1^ = 7.5 cm, t ^2^ = 0.2 cm)	Unspecified	Group 1: 1.7 mA Group 2: 1.2 mA unspecified time	Two stimulation sessions	Speech comprehension test (rhythmic noise-vocoded speech)	0–280 ms, in 40-ms steps	0–7π/4, in steps of π/4	Modulation of speech perception induced by tACS in the case of bilateral stimulation using ring electrodes
Kadir, et al., 2020 [105]	17	Envelope—tACS/sham	Simultaneous T7 and T8 (35 cm^2^)	Either side of the Cz (35 cm^2^)	between 0.2–1.5 mA unspecified time	Two stimulation sessions	Speech comprehension test (SRT assessment)	Fixed lag times of 100 ms and 250 ms	0–5π/3, in steps of π/3	Between a short latency of 100 ms and a long latency of 250 ms, the modulation of speech understanding by the phase of the stimulus was different
Keshavarzi, et al., 2020 [106]	18	Envelope—tACS/sham	Simultaneous T7 and T8 (35 cm^2^)	Either side of the Cz (35 cm^2^)	between 0.7 mA-1.3 mA unspecified time	Two stimulation sessions	Speech comprehension test (SRT assessment)	Unspecified	0–5π/3, in steps of π/3	The tACS effect speech understanding in the θ band, had no effect on speech understanding in the δ band Compared with the sham stimulation, the current stimulation in the θ band can improve the understanding of speech in noise

^1^ d, diameter; ^2^ t, thickness.

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
