# Peer review of "Effects of Transcranial Electrical Stimulation on Human Auditory Processing and Behavior—A Review"

_brainsci, 2020, doi:10.3390/brainsci10080531_

Round 1
Reviewer 1 Report
General comments: The review tackles the effects of noninvasive electrical brain stimulation on several aspects of auditory processes. The topic merits to be reviewed. However, this work has several limitations that potential limits its interest.
The manuscript requires extensive English editing by a native speaker. Several sentences are difficult to read (several examples are provided below), several employed terms are not frequently employed in tES research and merits to be clarified, and the authors alternate in using full words or abbreviations (minutes vs. min).
Instead of providing a synthesis of the studies focusing on each auditory process, the authors seem to list the summaries of the included studies.
Regarding research criteria, this reviews lacks the details about the keywords used to perform the research, the research time window (the research was performed from … to … 2020, and included studies published anytime in English or published from … to … ) a potential flowchart stating the study inclusion/exclusion, a specification on the included populations (healthy versus clinical), etc. (please refer to PRISMA Guideline).
Most importantly, in the actual form, the review seems to lack several works that warrant citation and discussion. Some of the available cited works merit to be further clarified.
Before reviewing the studies, a brief explanation of the neurophysiological measures (e.g., waves names, amplitudes, frequencies; sensory gating) is helpful for nonspecialists and naïve readers.
The rationale behind only including studies published after 2015 in the table is not clear and warrants an explanation.
Since this work is a review, it would be important to abide to the classical abstract section by including the research criteria, the number of studies included, and the level of evidence obtained.
Below are some examples for English editing and other specifications:
- “tES has been proved its value”, “the regulation of tES on auditory”
- At the end of the abstract, the authors mention that they will review “speech processes” which does not seem to be the initial objective of this review
- In the introduction, the first sentence requires a reference especially about nerve repairing and remodeling. The concept reported in the following sentence is not clear and requires paraphrasing : the excitability of anode neuron cells is enhanced to produce depolarization of resting membrane potential, and the decrease of cathode excitability produces hyperpolarization. Paraphrasing : and tES which is a non-invasive brain stimulation to regulate neural processes has been widely studied. Paraphrasing : the research of tES is still mainly focus on the sports [12] and vision [13]. In addition, regarding this sentence, to the best of the reviewer’s knowledge, tES did not mainly focus on sports or vision but includes several other applications. Maybe the authors wanted to say that the physiological effects of tDCS have been the focus of studies in fields of sports and vision.
- The title of section 2 needs to be modified (Electrophysiological effects of auditory cortex). IT should be something like ‘Physiological effects of tES …”
- The reporting of the study by Zaehle et al. could be improved. First, in l. 53. Saying active electrodes (negative / anode) is not correct since the active electrode was an anode in one condition and a cathode in another condition.
- Regarding Terada, the stimulation side needs to be specified (left), three stimulations should be three stimulation conditions, cathode tDCS should be cathodal tDCS, the reference electrode site should be specified as done in the previous study.
- The statement in l. 86-89 is premature. This could be spared for the discussion part.
- In the study by Boroda and colleagues : false stimulation? What are the electrodes positions according to the 10-20 EEG system for bilateral primary auditory cortex?
- In l. 105-106, the word target could be replaced with study ‘Therefore, tDCS can be used as an effective tool to target the plasticity of the auditory cortex’.
- In the study by Chen and colleagues, the stimulation site was not specified. Seveal things need to be clarified as well: four different tDCS “occasions” (l. 110)? non-tDCS? Stimulation interval (did the authors mean washout intervals)?
- Impey et al. research also “verified”.
Author Response
Thanks for your comment. According to the reviewer’s comment we provide a point-by-point response. Please see the attachment.

Reviewer 2 Report
Effects of Transcranial Electrical Stimulation on Humand Auditory Processing and Behavior – A review Summary:TES as a non-invasive technique which allows for neuromodulation of the brain holds a lot of potential for many applications. Among those, influencing human auditory processing is a growing field that only recently left its infancy and leaves yet a lot to explore. In recent years, studies have found electrophysiological effects of tES on auditory processing, e.g. modulation of auditory evoked potentials, changing the excitability of auditory neurons or entraining specific frequencies in the auditory cortex. In addition, studies further found behavioral effects of tES on auditory processes such as temporal resolution and attention as well as speech processing. But many questions are still left unanswered, requiring further efforts in this field to, hopefully, derive effective clinical treatments of hearing-related problems. The manuscript contributes a valuable oversight about past auditory-related tES studies. Nevertheless, I think it would be helpful, if not necessary, that a native speaker proofreads the text.
Major Concerns:
- Language; the grammar is far from optimal and not only makes it very difficult to read, but sometimes even obscures the meaning of sentences entirely.
- Structure of Review; while a broad structure is given via the sections and their subsections, the manuscript fails to integrate the referenced literature into a form that paints a clear picture about how, when and by whom progress was made in this field. Instead it reads as an unsorted collection of abstracts of studies. Each section should build on itself, beginning with earlier studies giving a general overview about a subject, leading to newer studies which incrementally added their own piece of knowledge to the subject. After reading each section, the reader needs to have a good understanding how and why the questions about the subject were raised, which answers were found to that and where the future may lead, listing concrete questions/problems that need yet to be answered/solved by future studies.
- I like Table 1 which gives a good overview about the referenced studies which are dealing with the same subject. Consider adding tables for section 3 and 4 as well, which would give a quick overview about the main studies discussed in these sections and allow for a quick comparison about tES parameters used in these studies.
- The description of Figure 1 gives no indication to which experiment it belongs or where it was taken from. In the text where figure 1 is referenced, it only says that it belongs to “The above experiment [...]” with study 81 being the study directly above this statement, while this study does not contain this figure.
- The manuscript is very detailed in describing the designs of each referenced study, to a point where an overabundance of information is given, where non-essential information bloats the text and makes it very difficult to extract key information. Consider reducing information that does not contribute to key findings of study (e.g. don't list number of trials, pause lengths between blocks etc.) but rather focus on leaving in information that was essential to answer the hypothesis (e.g. conditions, used paradigm). Also, as mentioned above, adding more tables like Table 1 and outsourcing information from text to table reduces bloating.
- On the other hand, the manuscript is not detailed enough concerning the explanation of key terms or paradigms of the referenced studies, giving often very little or no information about them. While a short explanation of terms typical tES terms may be sufficient, other terms that do not directly originate from the field of tES need more explanation.
Minor concerns:
- line 13: Grammar; remove the “been”
- line 18: “Regulation of tES”; consider replacing “regulation” by “effect” or “neuromodulatory effect” since in this context, “regulation” does not really fit
- line 25-28: Please add a source for each claimed tES benefit/effect.
- Line 31: Regarding “densities”: maybe consider using the word "intensities" here, since the currents themselves are not dense but rather differ in amplitude; only the size of the affected area could be described as dense (but even then the preferred term would be "focal")
- Line 32: Consider replacing "tES" with "tDCS", since this sentence describes only tDCS effects but not effects of the other tES methods
- Line 33-34: "Anode neuron cells" reads unclear; rather describe them as "neuron cells affected by the anode"
- Line 34:”[...] the excitability of anode neuron cells is enhanced to produce depolarization of resting membrane potential [...]“ worded a bit unclear, the point that the resting membrane potential is depolarized to facilitate the forming of action potentials is missing
- Line 34-35: Same as before; it reads not clear enough that the decrease of excitability of cathode stimulated neurons is because of membrane hyperpolarization
- Line 38: Regarding reference [7]: since you are describing the very foundation on which the tACS effect builds here, namely entrainment, maybe cite one of the earlier studies which came up with entrainment in the first place, rather than a modern study which is just an application of entrainment
- line 42: Consider adding an explanation for stochastic resonance
- line 51: Consider adding the word "stimulation" to the title
- line 53: Consider replacing "negative" with "cathode" since you wrote "anode" and not "positive", to keep nomenclature consistent
- line 55: In the text it is said that “Each subject received a continuous experiment four times a week [...].”, while the cited source describes it as "consecutive sessions at 1-week intervals", meaning: in the span of 4 consecutive weeks, each week contained 1 session. Therefore the experiment was done in a month with 1 session per week
- Line 55-56: “[...] in which the temporal region (anode T7 and cathode T7) was stimulated twice [...]”: makes it sound like an electrode with positive charge (anode) as well as an electrode with negative charge (cathode) were both stacked on T7 and stimulated twice with this setup. Consider reformulating to make it clear that those were two different conditions; one where the electrode on T7 carried a positive charge (anodal) and one condition where it carried a negative charge (cathodal)
- Line 57: Same as line 55-56
- Line 59: Consider adding the crucial information that it was not a continuous tone but rather "a sequence of tones varying in loudness", thereby allowing to analyse AEPs in the first place
- line 59-60: Consider adding one or two sentences explaining the P50 and N1 and to which part of auditory processing they correspond. Additionally, consider adding an explanation on how modulating the amplitude of these ERPs would imply a causal neuromodulatory effect of tDCS
- line 61: The "anode" is the electrode carrying the positive charge, the tDCS should rather be described as "anodal", meaning the to be stimulated area was stimulated with a positive current (same goes for cathodal tDCS)
- line 77: “[...] regulation of tDCS [...]”; consider replacing “regulation” by “effect” or “neuromodulatory effect” since in this context “regulation” does not really fit
- line 80: What kind of hearing task? Was it passive or active? What were subjects supposed to do in this task?
- line 81: Is this information really necessary to understand/explain the results of this study?
- Line 82: In the prior paragraph you used the term “N1”, while here you use the term “N100”. Since they are synonyms and to keep the nomenclature consistent consider using “N1” here as well.
- Line 82: Consider adding a short explanation of what the P200 is as well as how the P50, N1 and P200 would be modulated by tDCS and what a successful modulation means as a finding for the efficacy of tDCS
- Line 84: Consider replacing “stimuli” with “stimulation” since applying a continuous electrical current via tDCS is called a “stimulation” and not a “stimulus”
- Line 91: Consider adding an explanation of sensory tetanus.
- ...
- These given minor concerns hopefully serve as examples for systematic concerns that also affect the rest of the manuscript and lead to easier identifying and resolving of these concerns
Author Response

(The authors gave the same response as above.)

Round 2
Reviewer 1 Report
General comments:
The referee would like to thank the authors for the attention they paid to the raised concerns/comments. Despite the remarkable improvements following revision, several issues remain to be addressed.
- In general, the number of stimulation sessions need to be specified for each study in the text and/or in the table.
- Some sentences could be improved and many of them could be summarized to increase the fluidity of the manuscript. Example (l. 147-150): “The time point 45 s after the end of the ST stimulation was named T1. At the T1 point, the same sound stimulation as the baseline period was required. The time point 30 min after the end of the ST stimulation was named T2. At the T2 point, the same tone as the baseline tone was started, but only 90 sound stimulations lasting 6 min were presented. The entire experimental process was recorded by EEG for about 60 min.”
- Cathode and anode tDCS should be replaced with cathodal and anodal tDCS throughout the manuscript since this is what is conventionally used in tDCS literature.
- After reviewing all the studies in this field, some of which yielded positive physiological outcomes, the authors of this work started section 5 (the conclusion) by saying that “there is no definite conclusion about the mechanism of tES on the auditory cortex and its impact on auditory cognition, and tES studies in the field of hearing still lack an electrophysiological basis”. This could be moderated to be fair to the works that found significant effects.
- Below are some additional suggestions:
Section 1.
- 55 : only few studies have been performed the field of hearing. (English editing)
- 58: we reviews (English editing)
There seem to be an error in figure 1 (the boxes on the right side should be shifted up by 1 level; they should stem from the second and third boxes).
Section 2.
l.77 cathode and anode tDCS (should be cathodal and anodal tDCS)
l.80 Each subject received consecutive sessions (single session?)
The authors defined the acronym SG (standing for sensory gating) upon its first appearance, but then did not use it anymore in the text.
l.81-82 English editing: Of the four experiments, two received stimulation of the left temporal region, including one cathode stimulation and one anode stimulation
It should be specified when first referring to electrodes to state that it according to the international 10-20 EEG system of electrode positioning
- 82. cathode stimulation and one anode stimulation
- 85-86, The sham stimulus was performed 85 before the true stimulus
l.88 the acronym EEG needs to be defined upon its first appearance
- 160 : “with non-tDCS, …” would be replaced with : “without tDCS as well as with sham, anodal and cathodal tDCS”.
l.163: “the anodal or cathodal electrode” could be replaced with “the anode or the cathode”
The sentence in l. 180-181 lacks a reference.
- 187 (English editing): “However, Weigl et al. thought this might be related to the MMN was recorded only once in each session after each tDCS stimulation.”
- 243, the sentence lacks a verb: “Use MEG to measure the magnetic field caused by the electrical activity of neurons in the brain and perform spatial positioning”
Table 2/3 title should include a reference to the included data rather than “data of section 3/4”
Author Response
Dear reviewer,
Thanks for your comment. According to the reviewer’s comment, we have made some changes.
Please see the attachment.

Reviewer 2 Report
The authors made significant changes to style and language of the manuscript, therefore making it easier to read and understand. On top of that, they included new tables and a figure as well as reworking many aspects of the manuscript, making it much more informative and coherent.
Nevertheless, there are still some language errors to be found, making it necessary to further proof-read the text.
Examples:
- line 58: “reviews” needs to be changed to “review”
- figure 1: “with reasons that not including” is odd grammar and should be re-formulated
- figure 1: “with reasons the subjects not healthy human” odd grammar again
- line 68: “excluding duplicates 13 articles” needs to be changed, to e.g. “excluding 13 duplicate articles”
- line 75: “stereotyped” is the passive verb form while the adjective form would be “stereotypical”. This needs not necessarily be changed, because both make sense here, but it “felt odd” to me when reading, this may be due to personal taste though
- line 77: “cathode” and “anode” are generally used to describe the electrodes themselves, tDCS would rather be described as “cathodal” and “anodal”
- line 78: ”to which the term anodal or cathodal stimulation condition”: the sentence is not complete, consider adding a suitable word at the end, e.g. ”to which the term anodal or cathodal stimulation condition refers”. Alternatively you could delete this sentence entirely, because the fact that cathodal and anodal stimulation was used in this study has already been mentioned at line 77, which also implies that the active electrode was either anodal or cathodal depending on the condition.
- line 82: “cathode stimulation” and “anode stimulation” change to “cathodal stimulation” and “anodal stimulation”
- line 83: “cathode or anode” to “cathodal or anodal”
- line 84: “stimulation was” to “stimulations were”
- line 84: “This ensured that participants of each stimulation method participated” is odd grammar, consider changing e.g. to “This ensured that each participant received each stimulation method”
- line 85+86: “stimulus” needs to be changed to “stimulation”
- line 87: “dB”; since besides dBA there also exist other dB measures like e.g. dBSPL , consider changing to “dBA” to be more accurate. This might not be crucial though, since dBA is the typical measurement in this context anyway.
- ...
- These examples hopefully serve as a help to further improve the language quality of the manuscript as well as serving as motivation to further proof-read the text
If these concerns are adressed, a publication of the manuscript is warranted.
Author Response

(The authors gave the same response as above.)
